# REMOVING UNDESIRABLE FEATURE CONTRIBUTIONS USING OUT-OF-DISTRIBUTION DATA

**Saehyung Lee, Changhwa Park, Hyungyu Lee, Jihun Yi, Jonghyun Lee, Sungroh Yoon**[*]
Electrical and Computer Engineering, AIIS, ASRI, INMC, and Institute of Engineering Research
Seoul National University
Seoul 08826, South Korea
{halo8218,omega6464,rucy74,t080205,leejh9611,sryoon}@snu.ac.kr

## ABSTRACT

Several data augmentation methods deploy unlabeled-in-distribution (UID) data to bridge the gap between the training and inference of neural networks. However, these methods have clear limitations in terms of availability of UID data and dependence of algorithms on pseudo-labels. Herein, we propose a data augmentation method to improve generalization in both adversarial and standard learning by using out-of-distribution (OOD) data that are devoid of the abovementioned issues. We show how to improve generalization theoretically using OOD data in each learning scenario and complement our theoretical analysis with experiments on CIFAR-10, CIFAR-100, and a subset of ImageNet. The results indicate that undesirable features are shared even among image data that seem to have little correlation from a human point of view. We also present the advantages of the proposed method through comparison with other data augmentation methods, which can be used in the absence of UID data. Furthermore, we demonstrate that the proposed method can further improve the existing state-of-the-art adversarial training.

## 1 INTRODUCTION

The power of the enormous amount of data suggested by the empirical risk minimization (ERM) principle (Vapnik & Vapnik, 1998) has allowed deep neural networks (DNNs) to perform outstandingly on many tasks, including computer vision (Krizhevsky et al., 2012) and natural language processing (Hinton et al., 2012). However, most of the practical problems encountered by DNNs have high-dimensional input spaces, and nontrivial generalization errors arise owing to the curse of dimensionality (Bellman, 1961). Moreover, neural networks have been found to be easily deceived by adversarial perturbations with a high degree of confidence (Szegedy et al., 2013). Several studies (Goodfellow et al., 2014; Krizhevsky et al., 2012) have been conducted to address these generalization problems resulting from ERM. Most of them handled the generalization problems by extending the training distribution (Madry et al., 2017; Lee et al., 2020). Nevertheless, it has been demonstrated that more data are needed to achieve better generalization (Schmidt et al., 2018). Recent methods (Carmon et al., 2019; Xie et al., 2019) introduced unlabeled-in-distribution (UID) data to compensate for the lack of training samples. However, there are limitations associated with these methods. First, obtaining suitable UID data for selected classes is challenging. Second, when applying supervised learning methods on pseudo-labeled data, the effect of data augmentation depends heavily on the accuracy of the pseudo-label generator.

In our study, in order to break through the limitations outlined above, we propose an approach that promotes robust and standard generalization using out-of-distribution (OOD) data. Especially, motivated by previous studies demonstrating the existence of common adversarial space among different images or even datasets (Naseer et al., 2019; Poursaeed et al., 2018), we show that OOD data can be leveraged for adversarial learning. Likewise, if the OOD data share the same undesirable features as those of the in-distribution data in terms of standard generalization, they can be leveraged for standard learning. By definition, in this work, the classes of the OOD data differ from those of the in-distribution data, and our method do not use the label information of the OOD data. Therefore the

---

[*]Correspondence to: Sungroh Yoon sryoon@snu.ac.kr.

proposed method is free from the previously mentioned problems caused by UID data. We present a theoretical model which demonstrates how to improve generalization using OOD data in both adversarial and standard learning. In our theoretical model, we separate desirable and undesirable features and show how the training on OOD data, which shares undesirable features with in-distribution data, changes the weight values of the classifier. Based on the theoretical analysis, we introduce *out-of-distribution data augmented training* (OAT), which assigns a uniform distribution label to all the OOD data samples to remove the influence of undesirable features in adversarial and standard learning. In the proposed method, each batch is composed of training data and OOD data, and OOD data regularize the training so that only features that are strongly correlated with class labels are learned. We complement our theoretical findings with experiments on CIFAR-10 (Krizhevsky et al., 2009), CIFAR-100 (Krizhevsky et al., 2009), and a subset of ImageNet (Deng et al., 2009). In addition, we present the empirical evidence for the transferability of undesirable features through further studies on various datasets including Simpson Characters (Attia, 2018), Fashion Product Images (Aggarwal, 2018), SVHN (Netzer et al., 2011), Places365 (Zhou et al., 2017), and VisDA-17 (Peng et al., 2017).

**Contributions**  (i) We propose a simple method, *out-of-distribution data augmented training* (OAT), to leverage OOD data for adversarial and standard learning, and theoretical analyses demonstrate how our proposed method can improve robust and standard generalization. (ii) The results of experimental procedures on CIFAR-10, CIFAR-100, and a subset of ImageNet suggest that OAT can help reduce the generalization gap in adversarial and standard learning. (iii) By applying OAT using various OOD datasets, it is shown that undesirable features are shared among diverse image datasets. It is also demonstrated that OAT can effectively extend training distribution by comparison with other data augmentation methods that can be employed in the absence of UID data. (iv) The state-of-the-art adversarial training method using UID data is found to further improve by incorporating the proposed method of leveraging OOD data.

## 2 BACKGROUND

**Undesirable features in adversarial learning**  Tsipras et al. (2018) demonstrated the existence of a trade-off between standard accuracy and adversarial robustness with the distinction between robust and non-robust features. They showed the possibility that adversarial robustness is incompatible with standard accuracy by constructing a binary classification task that the data model consists of input-label pairs $(\boldsymbol{x}, y) \in \mathbb{R}^{d+1} \times \{\pm 1\}$ sampled from a distribution as follows:

$$y \stackrel{u.a.r}{\sim} \{-1, +1\}, \quad x_1 = \begin{cases} +y & \text{w.p. } p \\ -y & \text{w.p. } 1-p \end{cases}, \quad x_2, \ldots, x_{d+1} \stackrel{i.i.d.}{\sim} \mathcal{N}(\epsilon y, 1). \tag{1}$$

Here, $x_1$ is a robust feature that is strongly correlated with the label, and the other features $x_2, \ldots, x_{d+1}$ are non-robust features that are weakly correlated with the label. $\epsilon$ is small but sufficiently large such that a simple classifier attains a high standard accuracy, and $p \geq 0.5$. To characterize adversarial robustness, the definitions of expected standard loss $\beta_s$ and expected adversarial loss $\beta_a$ for a data distribution $D$ are defined as follows:

$$\beta_s = \mathop{\mathbb{E}}_{(\boldsymbol{x},y) \sim D} [\mathcal{L}(\boldsymbol{x}, y; \theta)], \quad \beta_a = \mathop{\mathbb{E}}_{(\boldsymbol{x},y) \sim D} \left[ \max_{\boldsymbol{\delta} \in S} \mathcal{L}(\boldsymbol{x} + \boldsymbol{\delta}, y; \theta) \right]. \tag{2}$$

Here, $\mathcal{L}(; \theta)$ is the loss function of the model, and $S$ represents the set of perturbations that the adversary can apply to deceive the model. For Equation (1), Tsipras et al. (2018) showed that the following classifier can yield a small expected standard loss:

$$f_{\text{avg}}(\boldsymbol{x}) = \text{sign}(\boldsymbol{w}_{\text{unif}}^\top \boldsymbol{x}), \quad \text{where } \boldsymbol{w}_{\text{unif}} = \left[ 0, \frac{1}{d}, \ldots, \frac{1}{d} \right]. \tag{3}$$

They also proved that the classifier is vulnerable to adversarial perturbations, and that adversarial training results in a classifier that assigns zero weight values to non-robust features.

**Transferability of Adversarial Perturbations**  Naseer et al. (2019) produced domain-agnostic adversarial perturbations, thereby showing common adversarial space among different datasets. They showed that an adversarial function trained on Paintings, Cartoons or Medical data can deceive the classifier on ImageNet data with a high success rate. The study findings show that even datasets from considerably different domains share non-robust features. Therefore a method for supplementing the data needed for adversarial training is presented herein.

**Undesirable features in standard learning**  Wang et al. (2020) noted that convolutional neural networks (CNN) can capture high-frequency components in images that are almost imperceptible to a human. This ability is thought to be closely related to the generalization behaviors of CNNs, especially the capacity in memorizing random labels. Several studies (Geirhos et al., 2018; Bahng et al., 2019) reported that CNNs are biased towards local image features, and the generalization performance can be improved by regularizing that bias. In this context, a method of regularizing undesirable feature contributions using OOD data is proposed, assuming that undesirable features arise from the bias of CNNs or insufficient training data and are widely distributed in the input space.

## 3  METHODS

### 3.1  THEORETICAL MOTIVATION

In this section, we analyze theoretically how OOD data can be used to make up for the insufficient training samples in adversarial training based on the dichotomy between robust and non-robust features. The theoretical motivation of using OOD data to reduce the contribution of undesirable features in standard learning can be found in Appendix B.

**Setup and overview**  We denote in-distribution data as target data. Given a target dataset $\{(\tilde{\boldsymbol{x}}^i, y^i)\}_{i=1}^n \subset \mathcal{X} \times \{\pm 1\}$ sampled from a data distribution $\tilde{D}$, where $\mathcal{X}$ is the input space, we suppose that a feature extractor $\Phi : \mathcal{X} \to \mathcal{Z} \subset \mathbb{R}^d$ and a linear classification model are trained on $\{(\tilde{\boldsymbol{x}}^i, y^i)\}_{i=1}^n$ and target feature-label pairs $\{(\Phi(\tilde{\boldsymbol{x}}^i), y^i)\}_{i=1}^n$, respectively, to yield the small expected standard loss of the classification model. We then define an OOD dataset $\{\hat{\boldsymbol{x}}^i\}_{i=1}^m \subset \mathcal{X}$ sampled from a data distribution $\hat{D}$ that has the same distribution of non-robust features as that of $\tilde{D}$ with reference to the preceding studies (Naseer et al., 2019; Moosavi-Dezfooli et al., 2017). After fixing $\Phi$ to facilitate the theoretical analysis on this framework, we demonstrate how adversarial training on the OOD dataset affects the weight values of our classifier.

**Our data model**  The feature extractor $\Phi$ can be considered to consist of several feature extractors $\phi : \mathcal{X} \to \mathcal{Z} \subset \mathbb{R}$. Hence, we can set the distributions of the target feature-label pair $(\Phi(\tilde{\boldsymbol{x}}), y) = (\tilde{\boldsymbol{z}}, y) \in \mathbb{R}^d \times \{\pm 1\}$ and the OOD feature vector $\Phi(\hat{\boldsymbol{x}}) = \hat{\boldsymbol{z}} \in \mathbb{R}^d$ as follows:

$$y \overset{u.a.r}{\sim} \{-1, +1\}, \quad \tilde{z}_1 \sim \mathcal{N}(y, u^2), \quad \tilde{z}_2, \ldots, \tilde{z}_{d+1} \overset{i.i.d.}{\sim} \mathcal{N}(\eta y, 1),$$
$$q \overset{u.a.r}{\sim} \{-1, +1\}, \quad \hat{z}_1 \sim \mathcal{N}(0, v^2), \quad \hat{z}_2, \ldots, \hat{z}_{d+1} \overset{i.i.d.}{\sim} \mathcal{N}(\eta q, 1),$$

(4)

where $u.a.r$ stands for uniformly at random. From here on, we will only deal with the OOD data, therefore the accents (tilde and caret) that distinguish between target data and OOD data are omitted. In Equation (4), the feature $z_1 = \phi_1(\boldsymbol{x})$ is the output of a robust feature extractor $\phi_1$, and the other features $z_2, \ldots, z_{d+1}$ are those of non-robust feature extractors $\phi_2, \ldots, \phi_{d+1}$. Since the OOD input vectors do not have the same robust features as the target input vectors, $z_1$ has zero mean and a small variance. Furthermore, because the OOD data have the same distribution of non-robust features as the target data, $z_2, \ldots, z_{d+1}$ have a non-zero mean and a larger variance than the robust features. In addition, $q$ represents the unknown label associated with the non-robust features, and $\eta$ is a non-negative constant which represents the degree of correlation between the non-robust features and the unknown label. Please note that the input space of our classifier is the output space of $\Phi$ in our data model. Therefore, $\eta$ is not limited to a small value even in the context of the $\ell_p$-bounded adversary. Rather, the high degree of confidence that DNNs show for the adversarial examples (Goodfellow et al., 2014) suggests that $\eta$ is large.

**Our linear classification model**  According to Section 2, we know that our linear classification model (logistic regression), defined as follows, yields a low expected standard loss while demonstrating high adversarial vulnerability.

$$p(y = +1 \mid \boldsymbol{z}) = \sigma(\boldsymbol{w}^\top \boldsymbol{z}), \; p(y = -1 \mid \boldsymbol{z}) = 1 - \sigma(\boldsymbol{w}^\top \boldsymbol{z}), \quad \text{where } \boldsymbol{w} = \left[0, \frac{1}{d}, \ldots, \frac{1}{d}\right]. \quad (5)$$

To observe the effect of adversarial training on the OOD dataset, we train our classifier by applying the stochastic gradient descent algorithm to the cross-entropy loss function $\mathcal{L}(; \boldsymbol{w})$.

Firstly, we construct the adversarial feature vector $\bar{z} = \Phi(x + \delta) : \delta \in S$ against our classifier for adversarial training.

**Theorem 1.** *Let* $t \in [0, 1]$ *be the given target value of the feature vector* $z$ *in our classification model, and* $\lambda$ *be a non-negative constant. Then, when* $t = 0.5$*, the expectation of the adversarial feature vector is*

$$\mathbb{E}_z[\bar{z}_1] = \mathbb{E}_z[z_1], \quad \mathbb{E}_z[\bar{z}_k] \approx \mathbb{E}_z[z_k] + \lambda \cdot q, \quad \text{where } k \in \{2, \dots, d+1\}. \tag{6}$$

(All the proofs of the theorems in this paper can be found in Appendix A.) Here, we assume the $\ell_\infty$-bounded adversary. In Theorem 1, we can observe that the adversary pushes the non-robust features farther in the direction of the unknown label $q$, which coincides with our intuition. When the given target value $t$ is 0.5, the adversary will make our classification model output equal to zero or one to yield a large loss.

Our classification model is trained on the adversarial features shown in Theorem 1.

**Theorem 2.** *When* $t = 0.5$*, the expected gradient of the loss function* $\mathcal{L}(\bar{z}, t; w)$ *with respect to the weight vector* $w$ *of our classification model is*

$$\mathbb{E}_{\bar{z}}\left[\frac{\partial \mathcal{L}}{\partial w_1}\right] \approx 0, \quad \mathbb{E}_{\bar{z}}\left[\frac{\partial \mathcal{L}}{\partial w_k}\right] \approx \frac{1}{2}(\eta + \lambda), \quad \text{where } k \in \{2, \dots, d+1\}. \tag{7}$$

Thus, the adversarial training with $t = 0.5$ on the OOD dataset leads to the weight values corresponding to the non-robust features converging to zero while preserving $w_1$ from the gradient update. This shows that we can reduce the impact of non-robust features using the OOD dataset. We, however, should not only reduce the influence of the non-robust features, but also improve the classification accuracy using the robust feature. This can be achieved through the adversarial training on the target dataset. Accordingly, we show the effect of the adversarial training on the OOD dataset when $w_1 > 0$ in our example.

**Theorem 3.** *When* $t = 0.5$ *and* $w_1 > 0$*, the expected gradient of the loss function* $\mathcal{L}(\bar{z}, t; w)$ *with respect to the weight vector* $w$ *of our classification model is*

$$\mathbb{E}_{\bar{z}}\left[\frac{\partial \mathcal{L}}{\partial w_1}\right] \approx \frac{1}{2}\lambda, \quad \mathbb{E}_{\bar{z}}\left[\frac{\partial \mathcal{L}}{\partial w_k}\right] \approx \frac{1}{2}(\eta + \lambda), \quad \text{where } k \in \{2, \dots, d+1\}. \tag{8}$$

Theorem 3 shows that when $w_1 > 0$, the adversarial training with $t = 0.5$ on the OOD dataset reduces the influence of all the features in $\mathcal{Z}$. However, we can see that the expected gradients for the weight values associated with the non-robust features are always greater than the expected gradient for the weight value associated with the robust feature. In addition, the greater the value of $\eta$, the faster the weight value associated with it converges to zero. This means that the contribution of non-robust features with high influence decreases rapidly. In the case of multiclass classification, it is straightforward that $t = 0.5$ corresponds to uniform distribution label $t_{\text{unif}} = [\frac{1}{c}, \dots, \frac{1}{c}]$, where $c$ is the number of the classes. Intuitively, the meaning of $(x, t_{\text{unif}})$ is that the input $x$ lies on the decision boundary. To reduce the training loss for $(x, t_{\text{unif}})$, the classifier will learn that the features of $x$ do not contribute to a specific class, which can be understood as removing the contributions of the features of $x$.

### 3.2 Out-of-distribution data augmented training

Based on our theoretical analysis, we introduce *Out-of-distribution data Augmented Training (OAT)*. OAT is the training on the union of the target dataset $\mathcal{D}_t$ and the OOD dataset $\mathcal{D}_o$. When applying our proposed method, we need to consider the following two points: 1) Temporary labels associated with OOD data samples are required for supervised learning, and 2) the loss functions corresponding to $\mathcal{D}_t$ and $\mathcal{D}_o$ should be properly combined.

First, we assign a uniform distribution label $t_{\text{unif}}$ to all the OOD data samples as confirmed in our theoretical analysis. This labeling method enables us to leverage OOD data for supervised learning at no extra cost. Moreover, it means that our method is completely free from the limitations of the methods using UID data (see Section 1).

Second, although OOD data can be used to improve the standard and robust generalization of neural networks, the training on target data is essential to enhance the classification accuracy of neural

networks. In addition, according to Theorem 3, adversarial training on the pairs of OOD data samples and $t_{\text{unif}}$ affects the weight for robust features as well as that for non-robust features. Hence, the balance between losses from $\mathcal{D}_t$ and $\mathcal{D}_o$ is important in OAT. For this reason, we introduce a hyperparameter $\alpha \in \mathbb{R}^+$ into our proposed method and train neural networks as follows:

$$
\begin{aligned}
\text{OAT-A:} \quad & \min_{\theta} \mathop{\mathbb{E}}_{(\boldsymbol{x}_t, y) \in \mathcal{D}_t} \left[ \max_{\boldsymbol{\delta} \in S} \mathcal{L}(\boldsymbol{x}_t + \boldsymbol{\delta}, y; \theta) \right] + \alpha \mathop{\mathbb{E}}_{\boldsymbol{x}_o \in \mathcal{D}_o} \left[ \max_{\boldsymbol{\epsilon} \in S} \mathcal{L}(\boldsymbol{x}_o + \boldsymbol{\epsilon}, t_{\text{unif}}; \theta) \right], \\
\text{OAT-S:} \quad & \min_{\theta} \mathop{\mathbb{E}}_{(\boldsymbol{x}_t, y) \in \mathcal{D}_t} \left[ \mathcal{L}(\boldsymbol{x}_t, y; \theta) \right] + \alpha \mathop{\mathbb{E}}_{\boldsymbol{x}_o \in \mathcal{D}_o} \left[ \mathcal{L}(\boldsymbol{x}_o, t_{\text{unif}}; \theta) \right].
\end{aligned}
\tag{9}
$$

Here, OAT-A and OAT-S represent OOD data augmented adversarial and standard learning, respectively. The pseudo-code for the overall procedure of our method is presented in Algorithm 1.

---

**Algorithm 1** Out-of-distribution augmented Training (OAT)

---

**Require:** Target dataset $\mathcal{D}_t$, OOD dataset $\mathcal{D}_o$, uniform distribution label $t_{\text{unif}}$, batch size $n$, training iterations $T$, learning rate $\tau$, hyperparameter $\alpha$, adversarial attack function $\mathcal{G}$
 1: **for** $t = 1$ **to** $T$ **do**
 2: $\quad (X_t, Y) = \text{SAMPLE}(\text{dataset} = \mathcal{D}_t, \text{size} = \frac{n}{2})$
 3: $\quad X_o = \text{SAMPLE}(\text{dataset} = \mathcal{D}_o, \text{size} = \frac{n}{2})$
 4: $\quad$ **if** Adversarial Learning **then**
 5: $\quad\quad \left[ \bar{X}_t, \bar{X}_o \right] \leftarrow \mathcal{G}([X_t, X_o], [Y, t_{\text{unif}}]; \boldsymbol{\theta})$
 6: $\quad$ **else if** Standard Learning **then**
 7: $\quad\quad \left[ \bar{X}_t, \bar{X}_o \right] \leftarrow [X_t, X_o]$
 8: $\quad$ **end if**
 9: $\quad$ *model update*:
10: $\quad\quad \boldsymbol{\theta} \leftarrow \boldsymbol{\theta} - \tau \cdot \nabla_{\theta} \text{AVERAGE}(\frac{1}{2} \mathcal{L}(\bar{X}_t, Y; \boldsymbol{\theta}) + \frac{\alpha}{2} \mathcal{L}(\bar{X}_o, t_{\text{unif}}; \boldsymbol{\theta}))$
11: **end for**
12: **Output:** trained model parameter $\boldsymbol{\theta}$

---

## 4 RELATED STUDIES

**Adversarial examples** Many adversarial attack methods have been proposed, including the projected gradient descent (PGD) and Carlini & Wagner (CW) attacks (Madry et al., 2017; Carlini & Wagner, 2017). The PGD attack employs an iterative procedure of the fast gradient sign method (FGSM) (Goodfellow et al., 2014) to find worst-case examples in which the training loss is maximized. CW attack finds adversarial examples using CW losses instead of cross-entropy losses. Recently, Croce & Hein (2020) proposed autoattack (AA), which is a powerful ensemble attack with two extensions of the PGD attack and two existing attacks (Croce & Hein, 2019; Andriushchenko et al., 2019). To defend against these adversarial attacks, various adversarial defense methods have been developed (Goodfellow et al., 2014; Kannan et al., 2018). Madry et al. (2017) introduced adversarial training that uses adversarial examples as training data, and Zhang et al. (2019) proposed TRADES to optimize a surrogate loss which is a sum of the natural error and boundary error.

**OOD detection** Lee et al. (2017) and Hendrycks et al. (2018) dealt with the overconfidence problem of the confidence score-based OOD detectors. They used uniform distribution labels as in our method to resolve the overconfidence issue. They did not, however, address the generalization problems of neural networks. Specifically, our theoretical results allow us to explain the classification performance improvement of classifiers that were only considered as secondary effects in the above-mentioned studies. Further related works can be found in Appendix C.

## 5 EXPERIMENTAL RESULTS AND DISCUSSION

### 5.1 EXPERIMENTAL SETUP

**OOD datasets** We created OOD datasets from the 80 Million Tiny Images dataset (Torralba et al., 2008) (80M-TI), using the work of Carmon et al. (2019) for CIFAR-10 and CIFAR-100, respectively. In addition, we resized (using a bilinear interpolation) ImageNet to dimensions of $64 \times 64$ and

Table 1: Accuracy (%) comparison of the OAT model with Standard, PGD, and TRADES on CIFAR10, CIFAR100, and ImgNet10 (64×64) under different threat models. We show the improved results compared to the counterpart of each model in bold.

| Model | Target | OOD | Clean | PGD100 | CW100 | AA |
|---|---|---|---|---|---|---|
| Standard | | - | 95.48 | 0.00 | 0.00 | 0.00 |
| PGD | | - | 87.48 | 49.92 | 50.80 | 48.29 |
| PGD+CutMix | CIFAR10 | - | 89.35 | 53.39 | 52.35 | 49.05 |
| TRADES | | - | 85.24 | 55.69 | 54.04 | 52.83 |
| $OAT_{PGD}$ | | 80M-TI | 86.63 | **56.77** | **52.38** | **49.98** |
| $OAT_{TRADES}$ | | 80M-TI | **86.76** | **59.66** | **55.71** | **54.63** |
| Standard | | - | 78.57 | 0.02 | 0.00 | 0.00 |
| PGD | | - | 61.37 | 24.66 | 24.68 | 22.76 |
| TRADES | CIFAR100 | - | 58.84 | 30.24 | 27.97 | 26.91 |
| $OAT_{PGD}$ | | 80M-TI | **61.54** | **30.02** | **27.85** | **25.36** |
| $OAT_{TRADES}$ | | 80M-TI | **63.07** | **34.23** | **29.02** | **27.83** |
| Standard | | - | 86.03 | 0.11 | 0.06 | 0.00 |
| PGD | ImgNet10 (64 x 64) | - | 82.80 | 48.77 | 48.86 | 48.34 |
| $OAT_{PGD}$ | | ImgNet990 | 81.91 | **59.03** | **54.69** | **53.83** |

$160 \times 160$ and divided it into datasets containing 10 and 990 classes, respectively; these are called ImgNet10 and ImgNet990. Furthermore, we resized Places365 and VisDA-17 for the experiments on ImgNet10 and cropped the Simpson Characters (Simpson), and Fashion Product (Fashion) datasets to dimensions of $32 \times 32$ for the experiments on CIFAR10 and CIFAR100. The details in sourcing the OOD datasets can be found in Appendix D.

**Implementation details**  Implementation details including the hyperparameter $\alpha$, architectures, batch sizes, and training iterations are summarized in Appendix E. All models compared in the same target dataset are trained in the same training batch size and iterations to ensure a fair comparison. In other words, OAT have a batch size of target $\frac{n}{2}$ + OOD $\frac{n}{2}$, which would be compared with a model that is normally trained with a batch size of $n$. To evaluate the adversarial robustness of the models in our experiments, we apply several adversarial attacks including PGD, CW, and AA. Note that we denote PGD and CW attacks with $T$ iterative steps as PGD$T$ and CW$T$, respectively, and the original test set as Clean. We compare the following models in our experiments[1]:

1. Standard: The model which is normally trained on the target dataset.
2. PGD: The model trained using PGD-based adversarial training on the target dataset.
3. TRADES: The model trained using TRADES on the target dataset.
4. $OAT_{PGD}$: The model which is adversarially trained with OAT based on a PGD approach.
5. $OAT_{TRADES}$: The model which is adversarially trained with OAT based on TRADES.
6. $OAT_{\mathcal{D}_o}$: The model which is normally trained with OAT using the OOD dataset $\mathcal{D}_o$.

## 5.2  STUDY ON THE EFFECTIVENESS OF OAT IN ADVERSARIAL LEARNING

Table 2: Comparison of OAT using various OOD datasets for improving robust generalization on CIFAR10 and ImgNet10 ($64 \times 64$). The None represents the baseline model (PGD).

| OOD | CIFAR10 | | | | ImgNet10 (64 x 64) | | |
|---|---|---|---|---|---|---|---|
| | None | SVHN | Simpson | Fashion | None | Places365 | VisDA17 |
| Clean | 87.48 | 86.16 | 86.79 | 85.84 | 82.80 | 82.37 | 82.46 |
| PGD20 | 50.41 | **53.70** | **53.88** | **53.27** | 49.00 | **59.86** | **55.34** |
| CW20 | 51.11 | **52.21** | **52.15** | **51.70** | 48.91 | **56.23** | **53.80** |

---

[1]Our codes are available at `https://github.com/Saehyung-Lee/OAT`.

**Evaluating adversarial robustness**   The improvements in the robust generalization performances of the PGD and TRADES models through the application of OAT are evaluated against PGD100, CW100, and AA with $\ell_\infty$-bound of $\frac{8}{255}$ and 0.031. The results are summarized in Table 1 and indicate that OAT improves the robust generalization of all adversarial training methods tested regardless of the target dataset. In particular, from the results against AA it can be seen that the effectiveness of OAT does not rely on obfuscated gradients (Athalye et al., 2018). This is because AA removes the possibility of gradient masking through the application of a combination of strong adaptive attacks (Croce & Hein, 2019; 2020) and a black-box attack (Andriushchenko et al., 2019).

However, while OAT brings a significant improvement in robustness against PGD attacks, it is relatively less effective against CW attacks. According to our theoretical analysis, these results imply that the OOD data have relatively few non-robust features used in the CW attacks. In other words, powerful targeted attacks, such as CW attacks, are constructed using gradient information more selectively than untargeted attacks, such as PGD attacks. To gain insight into this phenomenon, we train models with various hyperparameter $\alpha$ values. The results suggest that the greater the influence of OOD on the training process is, the higher the robustness against PGD attacks and the lower the robustness against CW attacks are (see Appendix G for more details).

In the absence of UID data, various data augmentation methods other than OAT can be employed. CutMix (Yun et al., 2019), a method of amplifying training data by cutting and pasting patches between training images, was recently proposed and exhibited excellent performance in classification and transfer learning tasks. By applying CutMix to adversarial training, the advantages of OAT over other data augmentation methods are assessed. Specifically, OAT brings a higher level of robustness than CutMix, as deduced from Table 1. Because adversarial examples are closely related to the high-frequency components of images (Wang et al., 2020) and adversarial training reduces the model's sensitivity to these components, CutMix is an ineffective data augmentation method. Although CutMix extends the global feature distribution, there is no significant difference in terms of local feature distribution. Contrarily, OAT can effectively regularize the classifier by observing various undesirable features held by additional data in the learning process.

**OAT with diverse OOD datasets**   Non-robust features are widely shared among different datasets, as demonstrated by applying OAT using various OOD datasets. Table 2 indicates that OAT improves robust generalization for all OOD datasets, including those that have little correlation with the target dataset from a human perspective. In addition, given that relatively simple datasets with no background (Fashion and VisDA-17) are less effective than others, we can suppose that non-robust features arise from the high complexity (Snodgrass & Vanderwart, 1980) of images for natural image datasets, such as CIFAR and ImageNet.

**When UID data are available**   In adversarial training, in-distribution data reduce the sensitivity of neural networks to non-robust features and provide robustly generalizable features. On the other hand, the proposed theory shows that OAT takes advantage of the data amplification effect only for non-robust features using OOD data. Therefore, the effectiveness of OAT is expected to decrease as more in-distribution data are augmented. To observe such a trend empirically, the effect of OAT is investigated according to additional in-distribution data size using previously published pseudo-labeled data (Carmon et al., 2019).

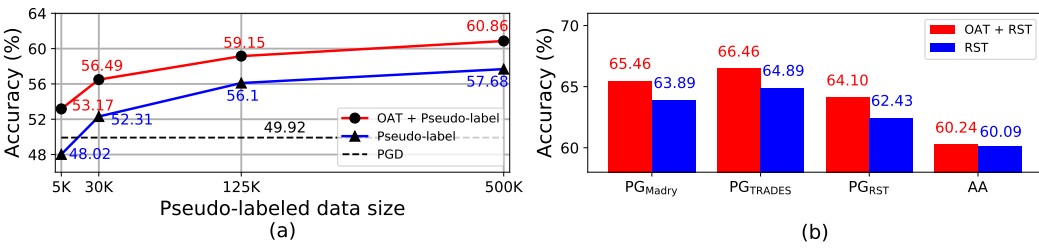

Figure 1: (a) The effectiveness of OAT with increasing pseudo-labeled data size under PGD100 and (b) the results of the state-of-the-art model improved by OAT under PG$_{TRADES}$, PG$_{Madry}$, PG$_{RST}$ (Carmon et al., 2019), and AA on CIFAR-10.

Surprisingly, Figure 1(a) shows that OAT still improves robust generalization even when many pseudo-labeled data are used. OAT is also combined with RST (Carmon et al., 2019), which has recently recorded a state-of-the-art adversarial robustness using UID data; in fact, Figure 1(b) demonstrates that OAT can further improve the state-of-the-art adversarial training method. This is presumably because the additional data include noisy labeled data, in which induce memorization in the learning process and thus impair the generalization performance (Zhang et al., 2016). OAT seems to achieve a higher level of robust generalization by effectively suppressing the effects of noisy data. Additional details on the experiments illustrated in Figure 1 are described in Appendix H.

### 5.3 STUDY ON THE EFFECTIVENESS OF OAT IN STANDARD LEARNING

**Randomization test** The effect of OAT is analyzed based on the randomization test (Zhang et al., 2016). The randomization test is an experiment aiming to observe the effective capacity of neural networks and the effect of regularization by training the model on a copy of the data where the true labels are replaced by random labels. In Figure 2(a), it can be seen that the Standard model

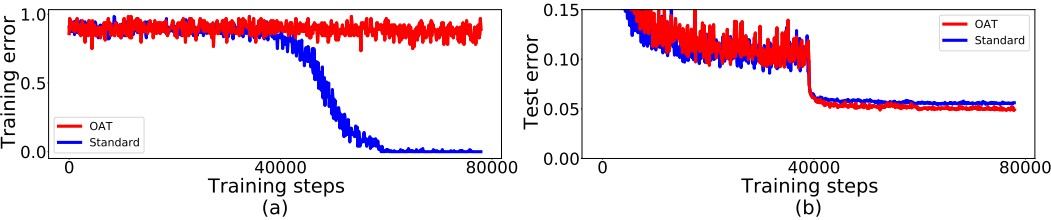

Figure 2: (a) The results of the randomization test and (b) the test error curves for the Standard and OAT models along their optimization trajectories on CIFAR-10.

memorizes all training samples to obtain a low training error, whereas the OAT model continues to have a high training error. These results show that OAT effectively regularizes neural networks so that it learns only features with strong correlation with the class labels. Figure 2(b) indicates that the OAT model learns more slowly than the Standard model owing to the influence of a strong regularizer in the early stages of training; however, it achieves a better generalization performance at the end of training.

**Evaluating classification performance** The effect of OAT on the classification accuracy is shown in Table 3. When the number of training samples ($N$) is small, the influence of undesirable features is expected to be large. Therefore, the experiments are classified depending on the number of training samples ($N$). Table 3 indicates that the effect of OAT is large when the amount of training data

Table 3: Accuracy (%, mean over 5 runs) comparison of the OAT model using various datasets with the baseline models. We show the improved results compared to the counterpart of each model in bold. Pseudo-label is the model trained using pseudo-labeled data (from 80M-TI) in conjunction with the target dataset. Fusion is the model that applies $OAT_{80M\text{-}TI}$ to the Pseudo-label model. A detailed description can be found in Appendix D.

| Dataset $N$ | CIFAR10 2,500 / Full | CIFAR100 2,500 / Full |
|---|---|---|
| Standard | 65.44 / 94.46 | 24.41 / 74.87 |
| $OAT_{SVHN}$ | **68.56** / 94.45 | **24.82 / 75.65** |
| $OAT_{Simpson}$ | **70.08** / 94.43 | **27.04 / 76.03** |
| $OAT_{80M\text{-}TI}$ | **72.49 / 95.20** | **26.13 / 76.30** |
| Pseudo-label | - / 95.28 | - / 77.24 |
| Fusion | - / **95.53** | - / **77.36** |

| Dataset $N$ | ImgNet10 (64 x 64) 100 / Full | ImgNet10 (160 x 160) 100 / Full |
|---|---|---|
| Standard | 37.90 / 86.93 | 33.36 / 90.91 |
| $OAT_{VisDA17}$ | 36.21 / 86.71 | **35.93 / 91.23** |
| $OAT_{Places365}$ | **41.84 / 88.37** | **40.11 / 91.42** |
| $OAT_{ImgNet990}$ | **42.18 / 87.88** | **40.41 / 91.87** |

is small, as predicted. Additionally, OAT enhances the generalization performance using 80M-TI, ImgNet990, and Places365, which have input distributions similar to the target datasets, more than when using other OOD datasets. This proves empirically that in our theoretical analysis, the OOD

data, following the same undesirable feature distribution as the target data, can improve generalization through OAT. In addition, the results of Pseudo-label and Fusion models show that even when pseudo-labeled data are available, OOD data can be leveraged to further improve the standard generalization performance. Moreover, the proposed method, which does not require complex operations and is very simple to implement, can lead to higher performance by combining with existing data augmentation methods. As an example, the effectiveness of Mixup (Zhang et al., 2017) is enhanced by applying OAT; the experimental results are provided in Appendix I.

Finally, OAT in a standard learning scheme using the entire target dataset is generally less effective than OAT in an adversarial training scheme (see Appendix F for more details). Therefore, it can be inferred that the transferability of undesirable features is greater in adversarial settings than in standard settings. In other words, these results experimentally show that the number of training samples required for robust generalization is large compared to that required for standard generalization.

## 6 CONCLUSIONS AND FUTURE DIRECTIONS

In this study, a method is proposed to compensate for the insufficient training data by using OOD data, which are less restrictive than UID data. It is theoretically demonstrated that training with OOD data can remove undesirable feature contributions in a simple Gaussian model. Experiments are performed on various OOD datasets, which surprisingly demonstrate that even OOD datasets that apparently have little correlation with the target dataset from the human perspective can help standard and robust generalization through the proposed method. These results imply that a common undesirable feature space exists among diverse datasets. In addition, the effectiveness of the proposed method is evaluated when extra UID data are available, and the results indicate that OAT can improve the generalization performance even when substantial pseudo-labeled data are used.

Nevertheless, some limitations need to be acknowledged. First, it is challenging to predict the effectiveness of the proposed method before applying it to a specific target-OOD dataset pair. Second, our method is less effective against strong targeted adversarial attacks, such as CW attacks, as it is difficult to generate deliberate adversarial attacks on in-distribution in the process of OAT. Therefore, as a future research direction, we aim to quantify the degree to which undesirable features are shared between the target and OOD datasets and construct strong adversarial attacks using OOD data.

**Acknowledgements:** This work was supported by the National Research Foundation of Korea (NRF) grant funded by the Korea government (Ministry of Science and ICT) [2018R1A2B3001628], the BK21 FOUR program of the Education and Research Program for Future ICT Pioneers, Seoul National University in 2020, and AIR Lab (AI Research Lab) in Hyundai & Kia Motor Company through HKMC-SNU AI Consortium Fund.

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

## A    PROOFS

**Theorem 1.** *Let $t \in [0,1]$ be the given target value of the feature vector $\boldsymbol{z}$ in our classification model, and $\lambda$ be a non-negative constant. Then, when $t = 0.5$, the expectation of the adversarial feature vector is*

$$\mathbb{E}_{\boldsymbol{z}}\left[\bar{z}_1\right] = \mathbb{E}_{\boldsymbol{z}}\left[z_1\right], \quad \mathbb{E}_{\boldsymbol{z}}\left[\bar{z}_k\right] \approx \mathbb{E}_{\boldsymbol{z}}\left[z_k\right] + \lambda \cdot q, \quad \text{where } k \in \{2, \ldots, d+1\}. \tag{10}$$

*Proof.* Let $\boldsymbol{\delta}_{\boldsymbol{x}}$ be the adversarial perturbation in the input space. Then,

$$\boldsymbol{\delta}_{\boldsymbol{x}} = \arg\max_{\boldsymbol{\delta}} \mathcal{L}(\Phi(\boldsymbol{x} + \boldsymbol{\delta}), t; \boldsymbol{w}) \approx \arg\max_{\boldsymbol{\delta}} \mathcal{L}(\Phi(\boldsymbol{x}) + \boldsymbol{\delta}^{\top}\nabla_{\boldsymbol{x}}\Phi, t; \boldsymbol{w})$$
$$= \arg\max_{\boldsymbol{\delta}} \mathcal{L}(\boldsymbol{z} + \boldsymbol{\delta}^{\top}\nabla_{\boldsymbol{x}}\boldsymbol{z}, t; \boldsymbol{w}) = \arg\max_{\boldsymbol{\delta}} \mathcal{L}(\boldsymbol{z} + \boldsymbol{\delta}_{\boldsymbol{z}}, t; \boldsymbol{w}), \quad \text{where } \boldsymbol{\delta}_{\boldsymbol{z}} = \boldsymbol{\delta}^{\top}\nabla_{\boldsymbol{x}}\boldsymbol{z}. \tag{11}$$

Equation 11 suggests that constructing the adversarial perturbation $\boldsymbol{\delta}_{\boldsymbol{x}}$ in the input space can be approximated by finding the adversarial perturbation $\boldsymbol{\delta}_{\boldsymbol{z}}$ in the feature space. These have a relationship of $\boldsymbol{\delta}_{\boldsymbol{z}} = \boldsymbol{\delta}_{\boldsymbol{x}}^{\top}\nabla_{\boldsymbol{x}}\boldsymbol{z}$ by linear approximation. Hence, without loss of generality, the perturbed feature vector $\bar{\boldsymbol{z}}$ can be approximated by $\boldsymbol{z} + \lambda \cdot \text{sign}(\nabla_{\boldsymbol{z}}\mathcal{L}(\boldsymbol{z}, t; \boldsymbol{w}))$ with an $\lambda > 0$. Then,

$$\mathbb{E}_{\boldsymbol{z}}\left[\bar{z}_1\right] = \mathbb{E}_{\boldsymbol{z}}\left[z_1 + \lambda \cdot \text{sign}(\nabla_{z_1}\mathcal{L}(\boldsymbol{z}, t; \boldsymbol{w}))\right] = \mathbb{E}_{\boldsymbol{z}}\left[z_1 + \lambda \cdot \text{sign}(w_1(\sigma(\boldsymbol{w}^{\top}\boldsymbol{z}) - \frac{1}{2}))\right]$$
$$= \mathbb{E}_{\boldsymbol{z}}\left[z_1\right] + \lambda\mathbb{E}_{\boldsymbol{z}}\left[\text{sign}(0)\right] = \mathbb{E}_{\boldsymbol{z}}\left[z_1\right]. \tag{12}$$

Because our classification model was trained to minimize the expected standard loss, $\mathbb{E}_{\boldsymbol{z}}\left[\text{sign}(w_k(\sigma(\boldsymbol{w}^{\top}\boldsymbol{z}) - \frac{1}{2}))\right]$ can be approximated by $q$. Then,

$$\mathbb{E}_{\boldsymbol{z}}\left[\bar{z}_k\right] = \mathbb{E}_{\boldsymbol{z}}\left[z_k + \lambda \cdot \text{sign}(\nabla_{z_k}\mathcal{L}(\boldsymbol{z}, t; \boldsymbol{w}))\right] = \mathbb{E}_{\boldsymbol{z}}\left[z_k + \lambda \cdot \text{sign}(w_k(\sigma(\boldsymbol{w}^{\top}\boldsymbol{z}) - \frac{1}{2}))\right]$$
$$\approx \mathbb{E}_{\boldsymbol{z}}\left[z_k\right] + \lambda \cdot q. \tag{13}$$

$\square$

**Theorem 2.** *When $t = 0.5$, the expected gradient of the loss function $\mathcal{L}(\bar{\boldsymbol{z}}, t; \boldsymbol{w})$ with respect to the weight vector $\boldsymbol{w}$ of our classification model is*

$$\mathbb{E}_{\bar{\boldsymbol{z}}}\left[\frac{\partial\mathcal{L}}{\partial w_1}\right] \approx 0, \quad \mathbb{E}_{\bar{\boldsymbol{z}}}\left[\frac{\partial\mathcal{L}}{\partial w_k}\right] \approx \frac{1}{2}(\eta + \lambda), \quad \text{where } k \in \{2, \ldots, d+1\}. \tag{14}$$

*Proof.* Based on the adversarial vulnerability of our classifier, $\sigma(\boldsymbol{w}^{\top}\bar{\boldsymbol{z}})$ can be approximated by $\frac{1}{2}(1+q)$. Therefore,

$$
\begin{aligned}
\mathbb{E}_{\bar{\boldsymbol{z}}}\left[\frac{\partial \mathcal{L}}{\partial w_1}\right] &= \mathbb{E}_{\bar{\boldsymbol{z}}}\left[\bar{z}_1(\sigma(\boldsymbol{w}^{\top}\bar{\boldsymbol{z}}) - \frac{1}{2})\right] = \mathbb{E}_{\bar{\boldsymbol{z}}}\left[\bar{z}_1 \cdot \sigma(\boldsymbol{w}^{\top}\bar{\boldsymbol{z}})\right] - \frac{1}{2}\mathbb{E}_{\bar{\boldsymbol{z}}}\left[\bar{z}_1\right] \\
&\approx \frac{1}{2}\mathbb{E}_{\bar{\boldsymbol{z}}}\left[\bar{z}_1\right](1+q) - \frac{1}{2}\mathbb{E}_{\bar{\boldsymbol{z}}}\left[\bar{z}_1\right] = \frac{q}{2}\mathbb{E}_{\bar{\boldsymbol{z}}}\left[\bar{z}_1\right] = 0, \\
\mathbb{E}_{\bar{\boldsymbol{z}}}\left[\frac{\partial \mathcal{L}}{\partial w_k}\right] &= \mathbb{E}_{\bar{\boldsymbol{z}}}\left[\bar{z}_k(\sigma(\boldsymbol{w}^{\top}\bar{\boldsymbol{z}}) - \frac{1}{2})\right] = \mathbb{E}_{\bar{\boldsymbol{z}}}\left[\bar{z}_k \cdot \sigma(\boldsymbol{w}^{\top}\bar{\boldsymbol{z}})\right] - \frac{1}{2}\mathbb{E}_{\bar{\boldsymbol{z}}}\left[\bar{z}_k\right] \\
&\approx \frac{1}{2}\mathbb{E}_{\bar{\boldsymbol{z}}}\left[\bar{z}_k\right](1+q) - \frac{1}{2}\mathbb{E}_{\bar{\boldsymbol{z}}}\left[\bar{z}_k\right] = \frac{q}{2}\mathbb{E}_{\bar{\boldsymbol{z}}}\left[\bar{z}_k\right] = \frac{1}{2}(\eta + \lambda).
\end{aligned}
\tag{15}
$$

$\square$

**Theorem 3.** *When $t = 0.5$ and $w_1 > 0$, the expected gradient of the loss function $\mathcal{L}(\bar{\boldsymbol{z}}, t; \boldsymbol{w})$ with respect to the weight vector $\boldsymbol{w}$ of our classification model is*

$$
\mathbb{E}_{\bar{\boldsymbol{z}}}\left[\frac{\partial \mathcal{L}}{\partial w_1}\right] \approx \frac{1}{2}\lambda, \quad \mathbb{E}_{\bar{\boldsymbol{z}}}\left[\frac{\partial \mathcal{L}}{\partial w_k}\right] \approx \frac{1}{2}(\eta + \lambda), \quad \text{where } k \in \{2, \ldots, d+1\}.
\tag{16}
$$

*Proof.*

$$
\begin{aligned}
\mathbb{E}_{\boldsymbol{z}}\left[\bar{z}_1\right] &= \mathbb{E}_{\boldsymbol{z}}\left[z_1 + \lambda \cdot \text{sign}(\nabla_{z_1}\mathcal{L}(\boldsymbol{z}, t; \boldsymbol{w}))\right] = \mathbb{E}_{\boldsymbol{z}}\left[z_1 + \lambda \cdot \text{sign}(w_1(\sigma(\boldsymbol{w}^{\top}\boldsymbol{z}) - \frac{1}{2}))\right] \\
&\approx \mathbb{E}_{\boldsymbol{z}}\left[z_1\right] + \lambda \cdot q, \\
\mathbb{E}_{\bar{\boldsymbol{z}}}\left[\frac{\partial \mathcal{L}}{\partial w_1}\right] &= \mathbb{E}_{\bar{\boldsymbol{z}}}\left[\bar{z}_1(\sigma(\boldsymbol{w}^{\top}\bar{\boldsymbol{z}}) - \frac{1}{2})\right] = \mathbb{E}_{\bar{\boldsymbol{z}}}\left[\bar{z}_1 \cdot \sigma(\boldsymbol{w}^{\top}\bar{\boldsymbol{z}})\right] - \frac{1}{2}\mathbb{E}_{\bar{\boldsymbol{z}}}\left[\bar{z}_1\right] \\
&\approx \frac{1}{2}\mathbb{E}_{\bar{\boldsymbol{z}}}\left[\bar{z}_1\right](1+q) - \frac{1}{2}\mathbb{E}_{\bar{\boldsymbol{z}}}\left[\bar{z}_1\right] = \frac{q}{2}\mathbb{E}_{\bar{\boldsymbol{z}}}\left[\bar{z}_1\right] = \frac{1}{2}\lambda.
\end{aligned}
\tag{17}
$$

$\square$

# B  THE THEORETICAL MOTIVATION OF OAT IN STANDARD LEARNING

Based on the same setup described in Section 3 of the main manuscript, we construct the data model for OAT in the following manner:

$$
q \overset{u.a.r}{\sim} \{-1, +1\}, \quad z_1 \sim \mathcal{N}(0, \sigma_1^2), \quad z_2 \sim \mathcal{N}(\kappa q, \sigma_2^2), \quad \text{where } \kappa \in \mathbb{R}^+, \ \sigma_1^2 \ll \sigma_2^2.
\tag{18}
$$

For simplicity, we consider only one desirable feature extractor $\phi_1$ and one undesirable feature extractor $\phi_2$. In Equation (18), features $z_1 = \phi_1(\boldsymbol{x})$ and $z_2$ represent the output of feature extractors $\phi_1$ and $\phi_2$, respectively. As the OOD input vectors do not have the same desirable features as the target input vectors, the mean of $z_1$ is zero. On the other hand, because the OOD data have the same distribution of undesirable feature as the target data, the mean of $z_2$ is non-zero. In addition, considering that the feature extractors are trained to respond sensitively to the target input vectors, the output variance of the undesirable feature extractor is much larger than that of the desirable feature extractor for OOD data, especially in the high-dimensional case. $q$ represents the unknown label associated with the undesirable feature, and $\kappa$ is a non-negative constant that represents the degree of correlation between the undesirable feature and the unknown label.

We can prove the following theorems for a logistic regression model that is not strongly dependent on the desirable feature.

**Theorem 4.** *The OOD data barely affect the gradient update of the weight associated with the desirable feature.*

*Proof.* Because we assumed a linear classifier that is not strongly dependent on the desirable feature owing to the bias of CNNs or insufficient training data, $\boldsymbol{w}^{\top}\boldsymbol{z} = w_1 z_1 + w_2 z_2$ can be approximated

by $w_2 z_2$. Then, for the target value $t$, we specified the feature $\boldsymbol{z}$ as shown in the following equation:

$$\mathbb{E}_{\boldsymbol{z}} \left[ \frac{\partial \mathcal{L}}{\partial w_1} \right] = \mathbb{E}_{\boldsymbol{z}} \left[ (\sigma(\boldsymbol{w}^\top \boldsymbol{z}) - t) z_1 \right]$$
$$\approx \mathbb{E}_{\boldsymbol{z}} \left[ (\sigma(w_2 z_2) - t) z_1 \right] = \mathbb{E}_{\boldsymbol{z}} \left[ \sigma(w_2 z_2) - t \right] \mathbb{E}_{\boldsymbol{z}} \left[ z_1 \right] = 0. \tag{19}$$

$\square$

Note that in Equation (19), the gradient of the loss function with respect to the weight value $w_1$ is zero regardless of $t$. The optimal $t$ is then a value that makes $w_2$ converge to zero, thereby removing the influence of the undesirable feature.

**Theorem 5.** *When $t = 0.5$, the standard learning on the OOD data leads to the weight value corresponding to the undesirable feature converging to 0.*

*Proof.*

$$\mathbb{E}_{\boldsymbol{z}} \left[ \frac{\partial \mathcal{L}}{\partial w_2} \right] = \mathbb{E}_{\boldsymbol{z}} \left[ (\sigma(\boldsymbol{w}^\top \boldsymbol{z}) - \frac{1}{2}) z_2 \right] \approx \mathbb{E}_{\boldsymbol{z}} \left[ (\sigma(w_2 z_2) - \frac{1}{2}) z_2 \right]. \tag{20}$$

When $w_2 > 0$, $z_2 \cdot \sigma(w_2 z_2) > \frac{1}{2} z_2$ with high probability. Hence,

$$\mathbb{E}_{\boldsymbol{z}} \left[ (\sigma(w_2 z_2) - \frac{1}{2}) z_2 \right] > 0, \quad \text{where } w_2 > 0. \tag{21}$$

Similarly,

$$\mathbb{E}_{\boldsymbol{z}} \left[ (\sigma(w_2 z_2) - \frac{1}{2}) z_2 \right] < 0, \quad \text{where } w_2 < 0. \tag{22}$$

$\square$

## C  FURTHER RELATED WORKS

**Using Unlabeled Data to Improve Adversarial Robustness**    Stanforth et al. (2019) and Carmon et al. (2019) analyzed the requirement of larger sample complexity for adversarially robust generalization (Schmidt et al., 2018). Based on a model described in a prior work (Schmidt et al., 2018), they theoretically proved that unlabeled data can alleviate the need for the large sample complexity of robust generalization. Therefore, they proposed a semi-supervised learning technique by augmenting the training dataset with extra unlabeled data. They used pseudo-labels obtained from a model trained with the existing training dataset. They experimented on the CIFAR-10 dataset by using the 80 Million Tiny Images dataset (Torralba et al., 2008) (80M-TI) as an extra training dataset, resulting in improvements of adversarial robustness of the model. Najafi et al. (2019) also used unlabeled data for adversarial robustness by extending the distributionally robust learning (Ben-Tal et al., 2013) to semi-supervised learning scenarios. Instead of using pseudo-labels, they used soft-labels, which are chosen from a set of labels and softened according to the loss values of data. Their results include experiments on the MNIST, CIFAR-10, and SVHN datasets.

**Comparison with Bad GAN**    Dai et al. (2017) theoretically showed that a perfect generator cannot able to enhance the generalization performance, and good semi-supervised learning actually requires a bad generator. Through theoretical analysis, they proposed an empirical formulation to generate samples with low-density in the input space. Compared to our method, there are two main differences between Bad GAN and OAT. First, Dai et al. (2017) only considered semi-supervised learning settings, whereas OAT can also be applied to adversarial settings. Second, data augmentation using GAN has clear limitations. Dai et al. (2017) penalized high-density samples to generate low-density samples in the input space, but this method is inefficient in a high-dimensional input space. On the other hand, OAT uses various and efficient OOD directly to regularize the model, and the research direction that develops from the use of synthetic data to the use of real data can also be found for OOD detection (Hendrycks et al., 2018).

Table 4: Implementation details for the experiments of OAT-A

| Target | architecture | $\alpha$ | training steps | batch size |
|--------|--------------|----------|----------------|------------|
| CIFAR | WRN-34-10 (Zagoruyko & Komodakis, 2016) | 1.0 | 80K | 128 |
| ImgNet10 | ResNet18 (He et al., 2016) | 1.0 | 15.4K | 128 |

Table 5: Implementation details for the experiments of OAT-S

| Target | N | architecture | $\alpha$ | training steps | batch size |
|--------|---|--------------|----------|----------------|------------|
| CIFAR | 2500 | ResNet18 | 1.0 | 4000 | 128 |
|  | 50K |  |  | 78K | 128 |
| ImgNet10 | 100 | WRN-22-10 | 1.0 | 200 | 100 |
| (64 x 64) | 9894 |  | 0.2 | 15.4K | 128 |
| ImgNet10 | 100 | ResNet18 | 1.0 | 400 | 100 |
| (160 x 160) | 9894 |  | 0.1 | 38.5K | 128 |

## D    SOURCING OOD DATASETS

We created OOD datasets from 80M-TI, using the work of Carmon et al. (2019) for CIFAR10 and CIFAR100, respectively. In other words, we trained a 11-way (1 more class for OOD) classifier on a training set consisting of the CIFAR10 dataset and 1M randomly sampled images from 80M-TI with keywords that did not appear in CIFAR10. We then applied the classifier on 80M-TI and sorted the images based on confidence in the OOD class. The 1M and 5M images selected in the order of the highest confidence were used for OAT-A and OAT-S, respectively. In Table 3, Fusion has a batch size of target $\frac{n}{2}$ + pseudo-labeled $\frac{n}{4}$ + OOD $\frac{n}{4}$, which would be compared with the Pseudo-label model that is trained with a batch size of target $\frac{n}{2}$ + pseudo-labeled $\frac{n}{2}$. In addition, we resized (using a bilinear interpolation) ImageNet to dimensions of $64 \times 64$ and $160 \times 160$ and divided it into datasets containing 10 and 990 classes, respectively; these are called ImgNet10 (train set size = 9894 and test set size = 3500) and ImgNet990, respectively. The classes were divided based on the Imagenette dataset (Howard), and the experimental results for the differently divided dataset (Imagewoof) can be seen in Appendix F. We increased the number of images of ImgNet990 by 10 times through random cropping and created the OOD datasets in the same process as 80M-TI. Furthermore, we resized (bilinear interpolation) Places365 (Zhou et al., 2017) and VisDA17 (Peng et al., 2017) for the experiments on ImgNet10 and cropped Simpson Characters (Simpson) (Attia, 2018) and Fashion product (Fashion) (Aggarwal, 2018) to dimensions of $32 \times 32$ for the experiments on CIFAR.

## E    IMPLEMENTATION DETAILS

For all the experiments except TRADES and OAT$_{TRADES}$, the initial learning rate is set to 0.1. The learning rate is multiplied by 0.1 at 50% and 75% of the total training steps, and the weight decay factor is set to 2e−4. We use the same adversarial perturbation budget $\epsilon = 8$, as in Madry et al. (2017). We recorded the maximum adversarial robustness of the models on the test set after the first learning rate decay in adversarial training and the empirical upper bound of the test accuracy of the models during the standard learning. The other details are summarized in Tables 4 and 5. For TRADES and OAT$_{TRADES}$, we deploy a batch size of 64 and train the models using the same configurations as Zhang et al. (2019).

## F    FURTHER DISCUSSION ABOUT THE EFFECTIVENESS OF OAT

We created a new ImgNet10 dataset with reference to Imagewoof (Howard) to learn more about the effects of OAT. Imagewoof is a subset of ImageNet, and all classes are dog breeds. We tested

Table 6: Accuracy (%) comparison of the OAT model with Standard and PGD on ImgNet10 (64×64) under different threat models. We show the improved results compared to the counterpart of each model in bold.

| Model | OOD | Clean | PGD20 | CW20 |
|---|---|---|---|---|
| Standard | - | 75.77 | - | - |
| PGD | - | 53.64 | 16.18 | 14.81 |
| OAT$_{PGD}$ | ImgNet990 | 48.24 | **22.09** | **17.1** |
| | Places365 | 50.45 | **22.57** | **17.37** |

Table 7: Accuracy (%, mean over 5 runs) comparison of the OAT models with the baseline models. We show the improved results compared to the counterpart of each model in bold.

| Target | N | 100 | 250 | 500 | 1250 | 2500 | Full |
|---|---|---|---|---|---|---|---|
| ImgNet10 (64 x 64) | Standard | 19.42 | 25.48 | 31.49 | 47.04 | 57.12 | 75.77 |
| | OAT$_{ImgNet990}$ | **22.74** | **30.07** | **35.16** | **48.84** | **58.14** | 75.17 |
| | OAT$_{Places365}$ | **22.36** | **30.15** | **33.57** | **48.16** | 57.15 | 75.63 |
| ImgNet10 (160 x 160) | Standard | 20.03 | 25.36 | 30.69 | 50.69 | 64.82 | 81.22 |
| | OAT$_{ImgNet990}$ | **23.64** | **30.05** | **37.09** | **60.01** | **70.39** | **83.16** |
| | OAT$_{Places365}$ | **23.09** | **31.74** | **37.22** | **59.45** | **69.90** | **83.04** |

whether OAT is effective for the newly constructed ImgNet10 dataset, and the experimental results of OAT-A and OAT-S are shown in Tables 6 and 7, respectively. We can see from Tables 6 and 7 that for the new ImgNet10, OAT is largely useful for generalization. However, from the results of ImgNet10(64×64) in Table 7, we can speculate that OAT will help in training as a regularizer only when an excessive number of features are involved in the classification. This is because OAT shows no performance improvement for the new ImgNet10(64×64) dataset. The new ImgNet10(64×64) is believed to have lost much of the categorical features, since all the classes are dog breeds, in addition to being overly downscaled. In other words, OAT is beneficial for the overfitting problem, but not for the underfitting problem. We can confirm this by artificially reducing the number of training samples to increase the features that can distinguish between each categorical sample distribution, and then observe the improvement in generalization performance by applying OAT.

## G ABLATION STUDY

Our experiments indicate that it is possible to enhance adversarial robustness by removing the contributions of non-robust features existing in additional data through OAT and transfer the consistency to in-distribution data. However, the increase in the robustness against targeted attacks appears smaller than that against untargeted attacks in our method. To gain insight into this phenomenon, we train the OAT models for various $\alpha$ values and investigate the difference between untargeted and targeted attacks. In Table 8, it can be seen that the greater the influence of OOD on the training process is, the higher the robustness against PGD20 and the lower the robustness against CW20 are. According to our theoretical analysis, it can be understood that there are many features used for untargeted attacks in OOD data, whereas relatively few features exist for targeted attacks. Because targeted attacks are

Table 8: Adversarial robustness (%) as hyperparameter $\alpha$ changes

| $\alpha$ | PGD20 | CW20 |
|---|---|---|
| 1.0 | 57.45 | 52.65 |
| 2.0 | 58.25 | 52.16 |
| 3.0 | 58.31 | 52.02 |
| 4.0 | 59.04 | 51.71 |
| 5.0 | 59.48 | 51.24 |

Table 9: Error rate (%) comparison of OAT with Mixup on CIFAR10, CIFAR100, and ImgNet10. We show the best result for each target dataset in bold.

| Model | Target | Ratio | Error rate |
|---|---|---|---|
| Standard | | - | 5.54 |
| $OAT_{80M-TI}$ | CIFAR10 | - | 4.80 |
| Mixup | | 0.0 | 4.11 |
| $OAT_{80M-TI}$+Mixup | | 0.6 | **3.88** |
| Standard | | - | 25.13 |
| $OAT_{80M-TI}$ | CIFAR100 | - | 24.35 |
| Mixup | | 0.0 | 22.63 |
| $OAT_{80M-TI}$+Mixup | | 0.6 | **21.60** |
| Standard | | - | 13.07 |
| $OAT_{ImgNet990}$ | | - | 12.12 |
| $OAT_{Places365}$ | ImgNet10 | - | 11.63 |
| Mixup | (64 x 64) | 0.0 | 11.60 |
| $OAT_{ImgNet990}$+Mixup | | 0.7 | 10.77 |
| $OAT_{Places365}$+Mixup | | 0.7 | **10.39** |
| Standard | | - | 9.09 |
| $OAT_{ImgNet990}$ | | - | 8.13 |
| $OAT_{Places365}$ | ImgNet10 | - | 8.58 |
| Mixup | (160 x 160) | 0.0 | 7.04 |
| $OAT_{ImgNet990}$+Mixup | | 0.3 | **6.46** |
| $OAT_{Places365}$+Mixup | | 0.3 | 6.90 |

stronger than untargeted attacks (Carlini & Wagner, 2017), the results of Table 8 provide empirical evidence for the trade-off between the transferability of adversarial perturbations and the strength of adversarial attacks.

A similar trend was reported in the study of Chan et al. (2020). The authors showed that input gradient adversarial matching (IGAM) can transfer robustness across different tasks. Their results show that IGAM-trained models have similar or higher robustness than baseline models against weak attacks, such as FGSM or low-step PGD attacks, but are vulnerable to strong attacks. The finetuning-based method can also be regarded as an attempt to increase the robustness by using other datasets, but the abovementioned phenomenon is not observed in the method proposed by Hendrycks et al. (2019). This is because the method does not transfer the robustness learned from other datasets to the target dataset but rather uses a function learned from a dataset that has a large sample complexity and a data distribution similar to that of the target data with little modification. This can be confirmed from the small number of training iterations and small learning rate involved by the fine-tuning process. Moreover, applying the same method to a dataset with a small sample size or far from the target data distribution has no effect (Chan et al., 2020).

## H WHEN UID DATA ARE AVAILABLE

We train the OAT models in conjuction with UID data as follows:

$$\min_{\theta} \alpha_{\text{in}} \mathbb{E}_{(\boldsymbol{x}_t,y)\in\mathcal{D}_t} \left[ \max_{\boldsymbol{\delta}\in S} \mathcal{L}(\boldsymbol{x}_t + \boldsymbol{\delta}, y; \theta) \right] + \alpha_{\text{o}} \mathbb{E}_{\boldsymbol{x}_o\in\mathcal{D}_o} \left[ \max_{\boldsymbol{\epsilon}\in S} \mathcal{L}(\boldsymbol{x}_o + \boldsymbol{\epsilon}, t_{\text{unif}}; \theta) \right]$$
$$+ \alpha_{\text{UID}} \mathbb{E}_{\boldsymbol{x}_u\in\mathcal{D}_{\text{UID}}} \left[ \max_{\boldsymbol{\zeta}\in S} \mathcal{L}(\boldsymbol{x}_u + \boldsymbol{\zeta}, y_{\text{pseudo}}; \theta) \right]. \tag{23}$$

In the experiment of Figure 1(a), the Pseudo-label models are trained via a PGD-based approach, and every batch consists of 64 CIFAR-10 data and 64 pseudo-labeled data. The OAT+Pseudo-label models are also trained via a PGD-based approach, and every batch consists of 64 CIFAR-10 data, 32 pseudo-labeled data, and 32 OOD data (80M-TI). The hyperparameters $(\alpha_{\text{in}}, \alpha_{\text{o}}, \alpha_{\text{UID}})$ are set to $(\frac{1}{3}, \frac{1}{3}, \frac{1}{3})$. All other conditions are the same as described in Appendix E. In the experiment of

Figure 1(b), we train the models with the same configurations as Carmon et al. (2019), but every batch for the OAT+RST model comprises 128 pseudo-labeled data, 64 CIFAR-10 data, and 64 OOD data (the RST model has a batch size of 256). The hyperparameters $(\alpha_{\text{in}}, \alpha_{\text{o}}, \alpha_{\text{UID}})$ are set to (0.25, 0.25, 0.5).

## I  EXPERIMENTAL RESULTS FOR MIXUP

Zhang et al. (2017) proposed a data augmentation method named Mixup. Mixup generates training examples as $\tilde{x} = \alpha x_i + (1 - \alpha)x_j$ and $\tilde{y} = \alpha y_i + (1 - \alpha)y_j$, where $(x_i, y_i)$ and $(x_j, y_j)$ are two examples drawn at random from the training data, and $\alpha \in [0, 1]$. Here, we show that even outside of the convex combination of training data can be effectively regularized using OOD data. Table 9 shows that the models trained with the combination of OAT and Mixup always output the highest accuracy. The OAT+Mixup models are trained with the following algorithm:

---

**Algorithm 2** OAT+Mixup

---

**Require:** Ratio $\gamma$, Target dataset $\mathcal{D}_t$, OOD dataset $\mathcal{D}_o$, uniform distribution label $t_{\text{unif}}$, batch size $n$, training iterations $T$, learning rate $\tau$

1: **for** $t = 1$ **to** $T$ **do**
2: $\quad (X_t, Y_t) = \text{SAMPLE}(\text{dataset} = \mathcal{D}_t, \text{size} = n)$
3: $\quad X_o = \text{SAMPLE}(\text{dataset} = \mathcal{D}_o, \text{size} = \lfloor n * \gamma \rfloor)$
4: $\quad (\bar{X}_t, \bar{Y}_t) \leftarrow \text{PERMUTE}(X_t, Y_t)$
5: $\quad (\bar{X}_t [0 : \lfloor n * \gamma \rfloor], \bar{Y}_t [0 : \lfloor n * \gamma \rfloor]) \leftarrow (X_o, t_{\text{unif}})$
6: $\quad (X, Y) \leftarrow \text{MIXUP}(X_t, Y_t, \bar{X}_t, \bar{Y}_t)$
7: $\quad$ *model update*:
8: $\quad \boldsymbol{\theta} \leftarrow \boldsymbol{\theta} - \tau \cdot \nabla_\theta \text{AVERAGE}(\mathcal{L}(X, Y; \boldsymbol{\theta}))$
9: **end for**
10: **Output:** trained model parameter $\boldsymbol{\theta}$

---

