# OpenReview forum: "Removing Undesirable Feature Contributions Using Out-of-Distribution Data"
_ICLR.cc/2021/Conference — ICLR 2021 Poster_

### Official Review · AnonReviewer1 · 2020-10-26

**Rating:** 6
**Confidence:** 4

**Review:**

Summary: This paper seeks to remove "non-robust" features by first hypothesizing that out-of-distribution image datasets share the same spurious features as the target data distribution, then performing adversarial training on the out-of-distribution data (against uninformative labels) to remove these spurious features. They apply the method to PGD and TRADES adversarial training and show empirically that it improves the clean and robust accuracies. When unlabeled in-distribution data is available, the method can be combined with robust self-training to get further gains.

Strengths:
- The method seems to be reasonably robust to the choice of OOD dataset, which makes the method quite flexible.
- The method gives consistant and substaintial gains beyond PGD and TRADES adversarial training, as well as robust self-training, when OOD data is used. This is useful particularly since robust self-training (RST) doesn't seem to work well with OOD data (filtering unlabeled TinyImages data to be more like CIFAR was crucial to RST's success).
- They test against a variety of attacks, including PGD with 100 steps, CW, and AA attacks.

Weaknesses: Given the generality of using OOD data, there should be some things to check to make sure the given intuitions are correct.

- To perhaps give a bit more supporting evidence that the OOD datasets help due to sharing the same non-robust features, some ablations could be done - for example, what if you use a synthetically generated OOD set (like Gaussian noise)? I think that random OOD examples could also help because it can provide a regularizing effect - recent works [1,2] show that generic regularization such as dropout or early stopping improve adversarial training methods. It's unclear to me that the benefit is necessarily from removing non-robust features.
- The toy theoretical model seems to rely on the fact that the OOD dataset specifically only contains nonzero mean features for non-robust components, essentially allowing the algorithm to differentiate the non-robust features and robust features. However, it's unlikely that real-world OOD datasets have \emph{only} the non-robust features.
- How much does the choice of OOD dataset matter - why was SVHN, Fashion etc used for CIFAR while Places, Visda only used for ImageNet? Does the method work for all pairs of datasets?
- If you combine all the OOD datasets, do we get even more gains? Why do we choose a particular OOD dataset rather than just using whatever images we can find?
- From Table 1 and Table 2, it seems that OAT tends to further degrade the clean accuracy of PGD, although it slightly improves the clean accuracy of TRADES.

Other:
- It's pretty unclear throughout what is mean by the notion of "undesirable features", which also seems to be used synonymously with "non-robust features".
- Can you use in-distribution unlabeled data with the OAT loss, and if so do we get a similar benefit?


[1] Yao-Yuan Yang, Cyrus Rashtchian, Hongyang Zhang, Ruslan Salakhutdinov, Kamalika Chaudhuri. A Closer Look at Accuracy vs. Robustness, 2020.
[2] Leslie Rice, Eric Wong, J. Zico Kolter. Overfitting in adversarially robust deep learning, 2020.

---

> ### Author Response · Authors · 2020-11-16
> **Response to Reviewer #1 (part 1/2)**
>
> We appreciate the reviewer's suggestion of experiments that could further elucidate the effects of the proposed method.
> ***
> $\qquad\qquad\qquad\qquad\qquad$__Table A__
>
> | $\quad\quad\quad$Model | $\quad$Clean$\quad$| $\quad$PGD100$\quad$ | $\quad$CW100$\quad$ |
> | :----------: | :-----: | :-----: | :-----: |
> | PGD | 87.48 | 49.92 | 50.80 |
> |$OAT_{PGD}$-noise| 86.37 | 51.45 | 51.25 |
> |$OAT_{PGD}$-ImageNet| 87.59 | 57.13 | 52.77 |
> |$OAT_{PGD}$-Places365| 87.75 | 57.49 | 51.89 |
> |$OAT_{PGD}$-UID| 86.86 | 56.28 | 52.06 |
> 1. __Q__: To perhaps give a bit more supporting evidence that the OOD datasets help due to sharing the same non-robust features, some ablations could be done - for example, what if you use a synthetically generated OOD set (like Gaussian noise)? I think that random OOD examples could also help because it can provide a regularizing effect - recent works show that generic regularization such as dropout or early stopping improve adversarial training methods. It's unclear to me that the benefit is necessarily from removing non-robust features.\
> __A__: We trained an $OAT_{PGD}$ model on CIFAR-10 using random noise and summarized the results in Table A ($OAT_{PGD}$-noise). The results indicate that the use of random noise can enhance adversarial robustness to some extent; however, it is relatively insignificant in comparison to the improvements observed in Table 1 and 2 within the manuscript. Therefore, the effect of OAT is not caused by generic regularization, but depends largely on (non-robust) features that are shared among different datasets.
> 2. __Q__: The toy theoretical model seems to rely on the fact that the OOD dataset specifically only contains nonzero mean features for non-robust components, essentially allowing the algorithm to differentiate the non-robust features and robust features. However, it's unlikely that real-world OOD datasets have only the non-robust features.\
> __A__: As you pointed out, various features reside in OOD images. However, the classifier in our theoretical model takes the output of the feature extractor as an input. This means that among the various features of the OOD images, only features that are useful in the in-distribution classification have a non-zero mean, because the features that are not useful in the classification will be filtered out while training the feature extractor. Therefore, only fragile (non-robust) features or robust features that are helpful to the classification will remain in the classifier input space.  We analyzed OAT under this assumption.
> Real-world OOD datasets may contain some robust features; however, since in-distribution data are being trained together, OAT will have a similar regularization effect to label-smoothing on robust features and will not cause performance degradation. For instance, OAT employing ImageNet [1], which partially shares robust features with CIFAR-10, leads to substantial improvement in adversarial robustness on the CIFAR-10 dataset (see $OAT_{PGD}$-ImageNet in Table A).
> 3. __Q__: How much does the choice of OOD dataset matter - why was SVHN, Fashion etc used for CIFAR while Places, Visda only used for ImageNet? Does the method work for all pairs of datasets?\
> __A__: Different datasets were used for CIFAR and ImageNet because they differ in dimension (32$\times$32 vs. 64$\times$64). Additionally, we experimented with a downsampled version of Places365 for the CIFAR-10 dataset. From the results in Table A, we can observe that OAT is robust to the choice of OOD dataset in the adversarial settings.

---

> > ### Author Response · Authors · 2020-11-16
> > **Response to Reviewer #1 (part 2/2)**
> >
> > 4. __Q__: If you combine all the OOD datasets, do we get even more gains? Why do we choose a particular OOD dataset rather than just using whatever images we can find?\
> > __A__: As shown in Table 1 and 2 within the manuscript, among OOD datasets, those close to in-distribution achieve a relatively large improvements in performance through OAT. We selected a specific OOD dataset to demonstrate this property of OAT. If we combine numerous OOD datasets, we can obtain more gains, but simply using more OOD datasets does not always result in a greater benefit against CW attacks that are constructed using gradient information more selectively than untargeted attacks. To show this, we trained an $OAT_{PGD}$ model on CIFAR-10 using a combination of SVHN, Simpson Characters, and Fashion Product Images. The evaluation results are shown in the table below. Comparing these results with Table 2 in the manuscript, it can be seen that the combination may result in slightly higher robustness against the PGD attack, but does not result in improved robustness against the CW attack.
> > |$\quad$Clean$\quad$| $\quad$PGD20$\quad$ | $\quad$CW20$\quad$ |
> > | :-----: | :-----: | :-----: |
> > | 85.92 | 53.97 | 52.11 |
> > 5. __Q__: From Table 1 and Table 2, it seems that OAT tends to further degrade the clean accuracy of PGD, although it slightly improves the clean accuracy of TRADES.\
> > __A__: Based on [2], we measured and reported the performance of the best checkpoint in our experiments.
> > In our implementation, the PGD and $OAT_{PGD}$ models have the highest robustness at the 50\% point of the total training epoch, whereas the TRADES and $OAT_{TRADES}$  models show the highest robustness at the 90\% point of the total training epoch.  This is the primary reason the $OAT_{PGD}$  and $OAT_{TRADES}$  models show different trends in terms of the clean accuracy. In other words, the same trend observed in Figure 2(b) in the manuscript appears in adversarial settings as well. In fact, $OAT_{PGD}$  leads to better clean accuracy than PGD in the later stages of training, as in $OAT_{TRADES}$.
> > 6. __Q__: It's pretty unclear throughout what is mean by the notion of "undesirable features", which also seems to be used synonymously with "non-robust features".\
> > __A__: As the effectiveness of OAT is investigated in both adversarial and standard settings in our study, we use a comprehensive word, "undesirable features". Therefore, "undesirable features" can be used synonymously with "non-robust features" in adversarial settings.
> > 7. __Q__: Can you use in-distribution unlabeled data with the OAT loss, and if so do we get a similar benefit?\
> > __A__: We trained an $OAT_{PGD}$ model on CIFAR-10 using UID data and summarized the results in Table A ($OAT_{PGD}$-UID). From our theoretical analysis, the effect of a uniform distribution label is to reduce the sensitivity of the classifier to all features that exist in the input images. Although, we exploit OOD data based on this knowledge, using in-distribution data for OAT does not necessarily impair adversarial robustness. If UID data is used as a complementary dataset, OAT removes the contributions of the non-robust features of the UID images. At the same time, OAT will have a similar effect to label-smoothing to robust features in training data (e.g., CIFAR-10), and information on newly available robust features will not be obtained from unlabeled data.
> > Therefore, OAT using UID data can lead to higher adversarial robustness than not using additional data, but shows limited effectiveness compared with pseudo-label-based methods, such as RST.
> >
> > [1] Chrabaszcz, Patryk, Ilya Loshchilov, and Frank Hutter. "A downsampled variant of imagenet as an alternative to the cifar datasets." arXiv preprint arXiv:1707.08819 (2017).\
> > [2] Rice, Leslie, Eric Wong, and J. Zico Kolter. "Overfitting in adversarially robust deep learning." arXiv preprint arXiv:2002.11569 (2020).

---

> > > ### Comment · AnonReviewer1 · 2020-11-22
> > > **Response to authors**
> > >
> > > Thanks to the authors for running the suggested experiments.
> > > The two strong points in this paper beyond the good empirical results are that the framework is quite flexible (the OOD datasets do not even need to have the same output space) and can potentially be more robust to dissimilar OOD datasets (though the gain is much worse) and can be combined with RST for better results. In my opinion, the motivation/theoretical model is a bit weak and somewhat disagrees with the empirical results (see below).
> > >
> > > The ablation with noise adds value to the method beyond naive OOD data (suggesting there is something special in the OOD data beyond regularization). The comparison of the performance of OAT on "similar" vs. "diverse" datasets is quite interesting, and I think a good understanding on these points could improve the paper, including adding the experiment about downsampling Places. One thing is, if we have a large OOD dataset but we're not sure if the whole thing is similar (for example, using TinyImages without doing the same filtering as in Carmon et al), whether we expect OAT will have an intermediate performance (thus giving some importance to filtering the dataset for similarity).
> > >
> > > However, I am still not convinced by the non-robust features view. Even though the theoretical model is on the deep features, as the authors mention, both non-robust and robust features are useful for in-distribution performance and will both be present in the features. I believe that this already differs from the assumptions in the paper - and I'm not sure what really breaks the symmetry in the effect on non-robust and robust features if all are present in OOD. The authors mention that for robust features, there is a regularizing effect - perhaps the intuition is that regularizing will cancel out the non-robust features more quickly than the robust ones. However, now I am confused why generic regularization, maybe something like label smoothing, would not do the same. There's also a bit of a conflict between this conceptual view and the experiments on similar and dissimilar OOD datasets - according to the theoretical model, shouldn't OOD datasets that are more dissimilar (don't have the same robust features, but has a bunch of non-robust features that are probably always present in natural images) be more effective?

---

> > > > ### Author Response · Authors · 2020-11-24
> > > > **Response to Reviewer #1**
> > > >
> > > > Thank you for the comments. Below, we would like to address the remaining concerns:
> > > > 1. __Q__: Even though the theoretical model is on the deep features, as the authors mention, both non-robust and robust features are useful for in-distribution performance and will both be present in the features. I believe that this already differs from the assumptions in the paper.\
> > > > __A__: We apologize for the confusion caused by not explicitly stating the assumptions about "real-world OOD datasets" in our first response. We consider the real-world OOD datasets as uncurated datasets with the potential to contain a subset of the (in-distribution) training set. Hence, the real-world OOD datasets may differ from the natural data model in our paper. This is because our theoretical model addresses an ideal case where no robust features are shared and all non-robust features are shared.
> > > >
> > > > 2. __Q__: I'm not sure what really breaks the symmetry in the effect on non-robust and robust features if all are present in OOD.\
> > > > __A__: First, based on the previous works on non-robust features [1,2], the adversarial vulnerability of standard classifiers indicates that CNNs are biased toward non-robust features. This bias makes the difference between the effect of OAT on robust features and that on non-robust features. In other words, the greater influence of non-robust features compared to that of robust features, greater effect on non-robust features than that on robust features in OAT. Second, half of the training mini-batch in OAT is occupied by the training (in-distribution) data. Adversarial training on the training data raises robust feature contributions and reduces non-robust feature contributions. This is another reason to break the symmetry in the effect on non-robust and robust features.
> > > >
> > > > 3. __Q__: I am confused why generic regularization, maybe something like label smoothing, would not do the same.\
> > > > __A__: Generic regularization techniques are inefficient in cases where selective regularization is required, such as in adversarial training, because they inhibit the entire learning ability of classifiers. In addition, they are not strong enough to eliminate undesirable feature contributions, as shown in a previous study [3]. In contrast, OAT is efficient because it is possible to regularize only the learning of features contained in OOD data, and as shown in Figure 2(a) in the main manuscript, it has a strong regularization effect that can completely suppress the learning of undesirable features.
> > > >
> > > > 4. __Q__: There's also a bit of a conflict between this conceptual view and the experiments on similar and dissimilar OOD datasets - according to the theoretical model, shouldn't OOD datasets that are more dissimilar (don't have the same robust features, but has a bunch of non-robust features that are probably always present in natural images) be more effective?\
> > > > __A__: If all OOD images share the same amount of non-robust features with the in-distribution data, as you mentioned, OOD datasets, which are more dissimilar, are more effective. However, our experiments on similar and dissimilar OOD datasets show that OOD datasets that are more similar to the in-distribution dataset share more non-robust features with the in-distribution dataset. This trend was also observed by Lee et al. [4], wherein the authors leveraged `boundary' samples (boundary of in-distribution) as the most effective samples for training confidence-calibrated classifiers. If it is assumed that the overconfidence problem for the OOD samples of classifiers is due to undesirable features, our experimental results are consistent with those reported by [4].
> > > >
> > > > [1] Tsipras, Dimitris, et al. "Robustness May Be at Odds with Accuracy." International Conference on Learning Representations. 2018.\
> > > > [2] Ilyas, Andrew, et al. "Adversarial examples are not bugs, they are features." Advances in Neural Information Processing Systems. 2019.\
> > > > [3] Zhang, Chiyuan, et al. "Understanding deep learning requires rethinking generalization." (2016).\
> > > > [4] Lee, Kimin, et al. "Training Confidence-calibrated Classifiers for Detecting Out-of-Distribution Samples." International Conference on Learning Representations. 2018.

---

### Official Review · AnonReviewer4 · 2020-10-28
**Neat idea! But still has a concern.**

**Rating:** 7
**Confidence:** 4

**Review:**

**1. Summary and contributions: Briefly summarize the paper and its contributions**
 In this work, authors looked at Out of Distribution (OOD) data from the data augmentation and regularization perspective and introduced Out-of-distribution data Augmented Training (OAT) based on their theoretical analysis which demonstrated that training with OOD data can remove undesirable feature contributions in a simple Gaussian model. The authors conducted experiments on both standard learning and adversarial learning and showed the effectiveness of OAT with strong results. Experimental results also imply that a common undesirable feature space exists among diverse datasets.

##########################################################################

**2. Strengths: Describe the strengths of the work. Typical criteria include: soundness of the claims (theoretical grounding, empirical evaluation), significance and novelty of the contribution, and relevance to the community.**

Very strong experimental results, it is clear from the results that OAT is effective for both standard learning and adversarial learning.

A sound theoretical analysis demonstrating that training with OOD data can remove undesirable feature contributions in a simple Gaussian model.

Neat randomization test that analyzed the effect of OAT for standard learning. It verified the claim that OAT regularizes the model to learn only features with a strong correlation with class labels, even though the generalization gap is rather small.

##########################################################################

**3. Weaknesses: Explain the limitations of this work along the same axes as above.**

In the middle of page 6: “In particular, from the results against AA it can be seen that the effectiveness of OAT does not rely on obfuscated gradients (Athalye et al., 2018).” It would be helpful to explain this a bit more.

Adversarial training on OOD data could be more clearly described before the theoretical analysis. How exactly are you using the OOD data? Maybe switch 3.1 and 3.2? Then it will be easier for the readers to follow the theoretical analysis. I think it will improve readability if Equation 9 is introduced earlier in the paper. Why not introduce OAT earlier?

For the rest of the writing issues, see the next section.
 ##########################################################################

**4. Clarity: Is the paper well written?**
Typos: \
Page 2: a high success rates\
Page 2: a human-an ability thought to be\
Ambiguity: \
Title: The word “remove” first appeared on page 4, the middle of the paper. Even though the paper conveyed the idea in the title, consider either change the title or more explicitly introducing the ideas in page 4 in the introduction section to improve readability. \
Page 3, The space X was never explicitly defined. \
Page 3, it’d be good to remind readers what is u.a.r\
Page 8, “can lead to higher performance by implementing existing data augmentation methods.” Did you mean when mixed with existing data augmentation methods?\
##########################################################################

**5. Reasons for score**
In conclusion, the ideas are very interesting and the strengths outweigh weakness by a large margin, so I would recommend accept.

---

> ### Author Response · Authors · 2020-11-16
> **Response to Reviewer #4**
>
> We appreciate the suggestions made by the reviewer to improve the readability of our paper.
> 1. __Q__: In the middle of page 6: "In particular, from the results against AA it can be seen that the effectiveness of OAT does not rely on obfuscated gradients (Athalye et al., 2018)." It would be helpful to explain this a bit more.\
> __A__: We added the following explanation of the relationship between obfuscated gradients and AA evaluation: "In particular, from the results against AA it can be seen that the effectiveness of OAT does not rely on obfuscated gradients (Athalye et al., 2018). This is because AA removes the possibility of gradient masking through the application of a combination of strong adaptive attacks (Croce & Hein, 2019; 2020) and a black-box attack (Andriushchenko et al., 2019)."
> 2. __Q__: Adversarial training on OOD data could be more clearly described before the theoretical analysis. How exactly are you using the OOD data? Maybe switch 3.1 and 3.2? Then it will be easier for the readers to follow the theoretical analysis. I think it will improve readability if Equation 9 is introduced earlier in the paper. Why not introduce OAT earlier?\
> __A__: To improve the readability of our paper, we added an explicit explanation of the proposed method in the introduction section as follows: "Based on the theoretical analysis, we introduce out-of-distribution data augmented training (OAT), which assigns a uniform distribution label to all the OOD data samples to remove the influence of undesirable features in adversarial and standard learning. In the proposed method, each batch is composed of training data and OOD data, and OOD data regularize the training so that only features that are strongly correlated with class labels are learned." In addition, we moved the pseudo-code for the proposed method in Appendix D to Section 3.2 to clearly describe the proposed method.
>
> We have also corrected typos and ambiguity found by the reviewer. Thank you again for the clarity check.

---

> > ### Comment · AnonReviewer4 · 2020-11-25
> > **Response to Authors**
> >
> > I have read the replies and the discussions, thanks for the changes. After reading the discussions between reviewer 1 and authors, I have decided to lower my rating from 8 to 7. This is still a good paper, but I would agree R1's concerns are valid, especially this one:
> >
> > **R1**: There's also a bit of a conflict between this conceptual view and the experiments on similar and dissimilar OOD datasets - according to the theoretical model, shouldn't OOD datasets that are more dissimilar (don't have the same robust features, but has a bunch of non-robust features that are probably always present in natural images) be more effective?
> >
> > **A**: If all OOD images share the same amount of non-robust features with the in-distribution data, as you mentioned, OOD datasets, which are more dissimilar, are more effective. However, our experiments on similar and dissimilar OOD datasets show that OOD datasets that are more similar to the in-distribution dataset share more non-robust features with the in-distribution dataset. This trend was also observed by Lee et al. [4], wherein the authors leveraged `boundary' samples (boundary of in-distribution) as the most effective samples for training confidence-calibrated classifiers. If it is assumed that the overconfidence problem for the OOD samples of classifiers is due to undesirable features, our experimental results are consistent with those reported by [4].
> >
> > I am not entirely convinced about "If it is assumed that the overconfidence problem for the OOD samples of classifiers is due to undesirable features" and I am not entirely convinced that "our experiments on similar and dissimilar OOD datasets show that OOD datasets that are more similar to the in-distribution dataset share more non-robust features with the in-distribution dataset. ".  How do you precisely measure "more non-robust features"? And if Reviewer 1 is right, then this shakes the foundation of the conceptual view proposed in this paper.

---

### Official Review · AnonReviewer3 · 2020-10-28
**Marginally below acceptance threshold**

**Rating:** 6
**Confidence:** 1

**Review:**

This paper proposes a new data augmentation method that utilizes out-of-distribution data for enhancing generalizability for both supervised and adversarial learning. While most of existing data augmentation methods explore auxiliary unlabeled in-distribution data, this paper tries to leverage out-of-distribution data for enhancing performance. Theoretical analysis is first presented and explains why out-of-distribution data can help. Then, a simple method motivated from the analysis is proposed and later verified by extensive experiments including both supervised and adversarial learning experiments.

The proposed method is technically sound and extensive experiments are conducted to verify the effectiveness of the proposed method in different datasets for both the supervised learning and adversarial learning tasks. The paper is well motivated and well written.

I am not familiar with the topic. My main concern is the technical contribution of this paper. It is straightforward that labeled data, though are out-of-domain, can enhance the capability of the feature extractor as the low layers of deep neural networks extract low-level semantics that are shared across images. There is no doubt about this. The paper provides some analysis on this and proposes a simple method which combines in-distribution and out-of-distribution data to train the model. The combination approach is very straightforward as well. I am not an expert in this area, but from an educated guess. I do not think this paper is good enough for acceptance.

---

> ### Author Response · Authors · 2020-11-16
> **Response to Reviewer #3**
>
> Thank you for the review. Below, we would like to address your main concerns:\
> __Q__:  It is straightforward that labeled data, though are out-of-domain, can enhance the capability of the feature extractor as the low layers of deep neural networks extract low-level semantics that are shared across images.\
> __A__: The motivation for multi-domain learning (MDL) [1,2] is to use low-level semantics that are distributed across different datasets to improve the performance of a given task. Hence, we present our contributions by comparing MDL with our method.
> 1. For MDL to achieve the desired effect, there must be a close relationship (from a human point of view) between the datasets used. This is because the effectiveness of MDL is obtained under the assumption that features shared between different datasets are beneficial for generalization. In contrast, in our study we assumed that the features shared between different datasets are detrimental to generalization, the desired effect is achieved by regularizing the contribution of those features. This effect is particularly important in adversarial training. This is because one of the main goals of adversarial training is to reduce the sensitivity of the classifier to non-robust features in the training dataset.
>
> 2. It is observed that features of interest in adversarial training have characteristics different from those of low-level semantics in the following phenomena. First, adversarial perturbations exhibit noise-like patterns, and when they are added to images, they drastically change the results of the classifiers without changing the semantics of the images.  Second, semantically different images share an adversarial space. For example, it is possible to construct adversarial perturbations that deceive the ImageNet classifier using Painting or Cartoons [3].  The concept of robust and non-robust features was introduced in previous studies to explain such phenomena that are incomprehensible from the standpoint of low-level semantics [4,5]. Based on the characteristics of the non-robust features mentioned above, we show that datasets that are significantly different from the main dataset can also be used to improve the performance on the main dataset in adversarial settings, unlike the case of MDL.
>
> [1] Caruana, Rich. "Multitask learning." Machine learning 28.1 (1997): 41-75.\
> [2] Dredze, Mark, Alex Kulesza, and Koby Crammer. "Multi-domain learning by confidence-weighted parameter combination." Machine Learning 79.1-2 (2010): 123-149.\
> [3] Naseer, Muhammad Muzammal, et al. "Cross-domain transferability of adversarial perturbations." Advances in Neural Information Processing Systems 32 (2019): 12905-12915.\
> [4] Tsipras, Dimitris, et al. "Robustness May Be at Odds with Accuracy." International Conference on Learning Representations. 2018.\
> [5] Ilyas, Andrew, et al. "Adversarial examples are not bugs, they are features." Advances in Neural Information Processing Systems. 2019.

---

### Official Review · AnonReviewer2 · 2020-10-31
**Official Blind Review #2**

**Rating:** 7
**Confidence:** 5

**Review:**

In this work, the authors propose to use out-of-distribution (OOD) data to improve the generalization of deep neural networks, especially against adversarial attacks. Theoretic analysis and experimental results demonstrate the effectiveness of such method. The idea is interesting and the paper is easy to follow. The experiments are also thorough and support their theoretic analysis.

However, I still have some concerns below:

1.	Unlabeled data [1,2,3] seems to be related to your work more than you claimed. And you missed two related works of unlabeled data. IPlease explain more on the difference between your method and previous works on unlabeled data and provide a more detailed comparison.

2.	You adopt OOD data and assign each data with uniform distribution label. I am curious on what would happen if your OOD data contain some data that is overlapped with the data in training set. Would it still work well?

[1] Yair Carmon, Aditi Raghunathan, Ludwig Schmidt, John C Duchi, and Percy S Liang. Unlabeled data improves adversarial robustness. In Advances in Neural Information Processing Systems, pp. 11190–11201, 2019.

[2] J. Uesato, J. Alayrac, P. Huang, R. Stanforth, A. Fawzi, and P. Kohli. Are labels required for improving adversarial robustness? In Advances in Neural Information Processing Systems (NeurIPS), 2019.

[3] Runtian Zhai, Tianle Cai, Di He, Chen Dan, Kun He, John E. Hopcroft, Liwei Wang. Adversarially Robust Generalization Just Requires More Unlabeled Data. arXiv Preprint arXiv: 1906.00555.

---

> ### Author Response · Authors · 2020-11-16
> **Response to Reviewer #2**
>
> Thank you for the reviewing the manuscript. Below, we would like to address your main concerns:
> 1. __Q__: Unlabeled data [1,2,3] seems to be related to your work more than you claimed. And you missed two related works of unlabeled data. IPlease explain more on the difference between your method and previous works on unlabeled data and provide a more detailed comparison.\
> __A__: All the previous works on unlabeled data [1,2,3] have proposed practically the same method.  The only difference among them is how they formulate the loss functions. Carmon et al. [1] used a surrogate loss introduced in [4], Uesato et al. [2] leveraged a combination of KL-divergence and cross-entropy loss, and Zhai et al. [3] used cross-entropy loss as in [5].  We have adopted RST [1] in our work as it brings the highest improvement in adversarial robustness among them.
> The main difference between previous studies and our study is the distribution of additional data leveraged. The previous studies used extra in-distribution data as additional data. For instance, Carmon et al. extracted and exploited in-distribution data of the CIFAR-10 dataset from 80 Million Tinyimages dataset (80M-TI) [6]. Uesato et al. and Zhai et al. divided CIFAR-10 into two subsets: one as labeled data and the other as unlabeled data. Their theoretical analysis also assumed that the unlabeled data were in-distribution, and when out-of-distribution data were used instead, a large performance drop can be observed. In contrast, we use OOD data as additional data. In our theoretical analysis, we assumed that the additional data were from OOD, and we showed the results of training with a uniform distribution label for those data. Because all datasets other than in-distribution for some selected classes can be used, OAT has a great advantage over methods that use unlabeled data [1,2,3] in terms of availability of additional data. Additionally, the previous works on unlabeled data rely heavily on the pseudo-label generator, but our method uses a uniform distribution label, leading to consistent performance improvement.
> 2. __Q__: You adopt OOD data and assign each data with uniform distribution label. I am curious on what would happen if your OOD data contain some data that is overlapped with the data in training set. Would it still work well?\
> __A__: If our OOD data contain some data that overlap with the data in the training set, OAT would be the same as applying a soft label to the overlapped training data. We trained an $OAT_{PGD}$ model on CIFAR-10 using a dataset containing OOD data (selected from 80M-TI) and CIFAR-10 in a one-to-one ratio. The evaluation results are shown in the table below. As the diversity of the OOD data decreased, the effectiveness of OAT on PGD100 decreased slightly compared to OAT using only 80M-TI, but still considerably improves the overall adversarial robustness.
> |$\quad$Clean$\quad$| $\quad$PGD100$\quad$ | $\quad$CW100$\quad$ |
> | :-----: | :-----: | :-----: |
> | 85.70 | 53.22 | 52.40 |
>
> [1] Carmon, Yair, et al. "Unlabeled data improves adversarial robustness." Advances in Neural Information Processing Systems. 2019.\
> [2] Uesato, Jonathan, et al. "Are labels required for improving adversarial robustness?." arXiv preprint arXiv:1905.13725 (2019).\
> [3] Zhai, Runtian, et al. "Adversarially robust generalization just requires more unlabeled data." arXiv preprint arXiv:1906.00555 (2019).\
> [4] Zhang, H., Yu, Y., Jiao, J., Xing, E., Ghaoui, L.E. & Jordan, M.. (2019). Theoretically Principled Trade-off between Robustness and Accuracy. Proceedings of the 36th International Conference on Machine Learning, in PMLR 97:7472-7482\
> [5] Madry, Aleksander, et al. "Towards Deep Learning Models Resistant to Adversarial Attacks." International Conference on Learning Representations. 2018.\
> [6] Torralba, Antonio, Rob Fergus, and William T. Freeman. "80 million tiny images: A large data set for nonparametric object and scene recognition." IEEE transactions on pattern analysis and machine intelligence 30.11 (2008): 1958-1970.

---

### Comment · ~Seungyong_Moon1 · 2021-06-15
**Missing related work**

Dear authors.

Nice to read this paper. The paper introduces standard and adversarial training methods that utilizes OOD data for improving standard or adversarial accuracies. However, a similar idea has been already proposed in [1] to improve adversarial accuracies and OOD detection performance. I think clearly stating the difference between [1] and yours would make your paper stronger.

[1] Maximilian Augustin, et. al., Adversarial Robustness on In- and Out-Distribution Improves Explainability, ECCV 2020

---

> ### Comment · ~Saehyung_Lee1 · 2021-06-15
> **Thank you!**
>
> Thank you for your interest in our work!
>
> As you commented, RATIO [1] seems very similar to our method (OAT-A). Actually, as we addressed in Section 4 of our paper, one or more previous studies that used uniform distribution labels as in our method can be found [2,3].
> However, one thing to note is that the primary goal of [1,2,3] is achieving reliability, and they considered the classification performance improvements (in standard or adversarial settings) as secondary effects.
> On the other hand, our primary goal is to achieve better generalization in both adversarial and standard settings, and thus, our theoretical and empirical analyses are focused on the generalization problems of neural networks, which are absent in the abovementioned studies.
>
> Again thank you for your valuable comments!
>
> [1] Augustin, Maximilian, Alexander Meinke, and Matthias Hein. "Adversarial robustness on in-and out-distribution improves explainability." ECCV, 2020.
>
> [2] Lee, Kimin, et al. "Training Confidence-calibrated Classifiers for Detecting Out-of-Distribution Samples." ICLR, 2018.
>
> [3] Hendrycks, Dan, Mantas Mazeika, and Thomas Dietterich. "Deep Anomaly Detection with Outlier Exposure." ICLR, 2018.

---

### Decision · Program_Chairs · 2021-01-07
**Final Decision**

**Decision:**

Accept (Poster)

**Comment:**

This paper studies the effect of using unlabelled out-of-distribution (OOD) data in the training procedure to improve robust (and standard) accuracies. The main algorithmic contribution is a data-augmentation based robust training algorithm to train a loss which is carefully designed to benefit from the additional OOD data. What's also interesting is that the OOD data is fed with random labels to the training procedure. As demonstrated in the theoretical results, this way of feeding OOD data helps to remove the dependency to non-robust features and hence improves robustness.

As pointed out by all the reviewers (which I agree with), the idea of using unlabelled OOD data at training is novel/interesting, and the paper also shows how this can be done algorithmically. The numerical results also confirm the effectiveness of the proposed methods.